# Divergent Directionality of Immune Cell-Specific Protein Expression between Bipolar Lithium Responders and Non-Responders Revealed by Enhanced Flow Cytometry

**DOI:** 10.3390/medicina59010120

**Published:** 2023-01-07

**Authors:** Keming Gao, Nicholas M. Kaye, Marzieh Ayati, Mehmet Koyuturk, Joseph R. Calabrese, Eric Christian, Hillard M. Lazarus, David Kaplan

**Affiliations:** 1Department of Psychiatry, University Hospitals Cleveland Medical Center, Cleveland, OH 44106, USA; 2Case Western Reserve University School of Medicine, Cleveland, OH 44106, USA; 3CellPrint Biotechnology, Cleveland, OH 44106, USA; 4Department of Computer Science, University of Texas Rio Grande Valley, Edinburg, TX 78539, USA; 5Department of Computer and Data Sciences, Center for Proteomics and Bioinformatics, Case Western Reserve University, Cleveland, OH 44106, USA; 6Department of Medicine-Hematology/Oncology, University Hospitals Cleveland Medical Center, Cleveland, OH 44106, USA

**Keywords:** lithium treatment, bipolar disorder, monocytes and CD4^+^ lymphocytes, biomarkers, intracellular proteins

## Abstract

*Background and Objectives*: There is no biomarker to predict lithium response. This study used *CellPrint*™ enhanced flow cytometry to study 28 proteins representing a spectrum of cellular pathways in monocytes and CD4^+^ lymphocytes before and after lithium treatment in patients with bipolar disorder (BD). *Materials and Methods*: Symptomatic patients with BD type I or II received lithium (serum level ≥ 0.6 mEq/L) for 16 weeks. Patients were assessed with standard rating scales and divided into two groups, responders (≥50% improvement from baseline) and non-responders. Twenty-eight intracellular proteins in CD4^+^ lymphocytes and monocytes were analyzed with *CellPrint*™, an enhanced flow cytometry procedure. Data were analyzed for differences in protein expression levels. *Results*: The intent-to-treat sample included 13 lithium-responders (12 blood samples before treatment and 9 after treatment) and 11 lithium-non-responders (11 blood samples before treatment and 4 after treatment). No significant differences in expression between the groups was observed prior to lithium treatment. After treatment, the majority of analytes increased expression in responders and decreased expression in non-responders. Significant increases were seen for PDEB4 and NR3C1 in responders. A significant decrease was seen for NR3C1 in non-responders. *Conclusions*: Lithium induced divergent directionality of protein expression depending on the whether the patient was a responder or non-responder, elucidating molecular characteristics of lithium responsiveness. A subsequent study with a larger sample size is warranted.

## 1. Introduction

In acute and maintenance treatment of bipolar disorder (BD), lithium is still a first-line medication [1]. About 1/3 to 2/3 of patients may reach treatment response, defined as ≥50% symptom improvement from baseline, in the acute phase of treatment [2,3,4,5]. However, the mechanism of lithium treatment response remains unclear [6] and there is no reliable predictor for lithium treatment response. Demographic and clinical characteristics have been compared between lithium responders and non-responders [3,7], and some of them have been considered as guidance in clinical practice. Along with its requirement for repeated laboratory monitoring of renal and thyroid functions, the use of lithium in BD continues declining [8] although lithium has unique neuroprotective and anti-suicidal effects [9,10]. However, if there is a biomarker or a panel of biomarkers for predicting lithium response, the use of lithium can be maximized, and the inconvenience and potential side effects related to lithium use can be avoided.

The effort of searching for biomarkers to predict lithium treatment response has been ongoing for decades. Using blood samples, researchers have investigated potential predictors for lithium response at different levels including genomic [11,12,13,14], gene expression [15,16,17,18,19,20], protein levels [21,22,23,24,25], neurotransmitters, signal transduction and pathways, endocrine systems, cytokines and immune systems, circadian rhythm, and mitochondria [6]. Brain imaging and brain activity-related measures have also been used to study the predictors for lithium treatment response [6,26,27]. However, these efforts have yet to produce reliable predictors of lithium responsiveness [28,29] although a large study from the International Consortium on Lithium Genetics (ConLi*Gen) found that bipolar patients with low genetic loading for schizophrenia had better response to lithium than those with high genetic loading [14].

However, studies from human induced pluripotent stem cells (iPSC) and lymphoblastoid cell lines found that lithium responders and non-responders had different molecular biomarkers [30,31,32]. Lithium treatment was linked to differential gene expression and different electrophysiological activities [30,31,32]. More importantly, only neurons derived from lithium-responders responded to lithium [30,31], and neurons derived from lithium non-responders responded to lamotrigine, but not to lithium [30]. In addition, differential gene expression and protein phosphorylation in those two types of neurons were also observed [32]. Studies from the Pharmacogenomic Study of Bipolar Disorder (PGDB) found that lithium response was related to the architecture of circadian rhythms and lithium treatment stabilized circadian disruptions [33,34]. These data suggest that bipolar lithium responders and non-responders can be separated with identifiable biomarkers that may be used in routine clinical practice. However, a very small contribution of each gene SNP to a complex disease, a mismatch between mRNA and protein levels [35,36,37], and the inability to measure post translational protein modifications, such as phosphorylation or methylation events have challenged the usefulness of genomic and transcriptomic approaches in studying biomarkers for predicting treatment response of complex diseases like BD. 

On the other hand, expressed protein levels and phosphorylation are highly correlated with cellular functions and phenotypes. Proteomic studies are not only likely to help us find different phenotypes based on lithium response and biomarkers, but also help us understand the pathology of BD. However, most plasma/serum-based technologies simultaneously interrogate the averaged productive capabilities of all cells in the body and thus are obligatorily low resolution and low sensitivity. Also, lithium can reach many organs, tissues, neurons, and non-neuronal cells, and act on different genes and pathways [6]. Therefore, it will be difficult to use low sensitive technologies to measure multiple proteins simultaneously in plasma/serum to find predictor(s) for lithium treatment response.

Flow cytometry can measure expression of multiple intracellular or surface-bound proteins including protein modifications in individual cells. This technology has been used in previous studies of BD [38,39,40,41,42]. However, flow cytometry has traditionally suffered poor signal to noise when measuring multiple markers at the same time or markers with low expression [43,44]. Enhanced flow cytometry developed by CellPrint Biotechnology, LLC (*CellPrint*™) is an innovative tyramide-based catalytic deposition labeling procedure, which improves dynamic range by 20 fold and signal to noise ratios by 10–100 fold compared to standard flow cytometric methods. The technology improves the capability of flow cytometers to report expression levels of low abundance analytes and intracellular molecules. As a result, this approach enables the detection and quantitative assessment of a wide variety of surface and intracellular proteins from numerous cell types [45,46,47,48,49,50,51,52]. In our previous analysis of 17 intracellular analytes, *CellPrint*™ was able to identify intracellular proteins in CD4^+^ lymphocytes and monocytes to differentiate bipolar lithium responders from non-responders [45]. The aim of this analysis was to assess differences in protein expression pre- and post- lithium for treatment responders and non-responders.

## 2. Methods

### 2.1. Study Design

The study design, study procedures, the inclusion and exclusion criteria, diagnosis, efficacy and safety assessments, and blood sample collection were detailed previously [45]. Briefly, this study was a 16-week open-label study of lithium monotherapy treatment of patients with BD type I or II who were at any phase of the illness and with at least mild symptoms (clinicaltrial.gov, NCT02909504). The diagnoses were ascertained with the MINI for DSM-5 and a structured research diagnostic interview. Standardized rating scales for depression, anxiety or mania were used for measuring symptom severity. Disability was measured with Sheehan Disability Rating scale and quality of life was measured with the Quality of Life, Enjoyment and Satisfaction Questionnaire. Rating scales were completed at baseline, week 1, 2, 4, 6, 8, 12, and 16. Eligible patients were treated with lithium for up to 16 weeks and ongoing unpermitted medications were tapered off by week 4. Blood samples of all patients were collected at the baseline and at the end of the study. Levels of intracellular proteins before and after lithium in monocytes and CD4^+^ lymphocytes of the lithium responders and lithium non-responders were measured.

### 2.2. Rationale of Using Monocytes and CD4^+^ Lymphocytes

The comparability of blood and brain have been investigated at different levels. At the DNA methylation level, the brain and blood are highly correlated. At the transcriptome level, whole blood [53,54] and peripheral blood mononuclear cells [55] had similar gene expression patterns as the brain tissues [56]. A systematic review has shown that neurotropic factors have similar changes in both central nervous system and peripheral blood system [25]. In addition, functional connections between brain and blood cells are through immune cells in the peripheral circulation with the brain [57,58].

Peripheral blood mononuclear cells (PBMC) have been used for genomic, genetic, gene expression, and protein expression studies of lithium treatment response in BD [16,59,60,61] as well as diagnostic and pathologic studies of BD [62,63,64,65,66,67]. Among the blood mononuclear cells, lymphocytes [38,68,69] and monocytes [69,70] have been extensively studied with flow cytometry for different purposes. In addition, collection of blood samples is relatively easy and cheap, and results from the PBMCs can be easily applied to routine clinical practice. Therefore, we chose monocytes and CD4^+^ lymphocytes as reporters of flow cytometry analysis in the current study.

### 2.3. Antibodies and Cytometric Analyses

*CellPrint*™ enhanced signal is generated by catalyzed reporter deposition and was used to measure the levels of intracellular protein in the study. The details of the procedure have been published elsewhere [45,46,47,48,49,50,51,52]. Briefly, antibodies targeting surface antigens to demarcate cell subtypes (CD4^+^ lymphocytes and monocytes) are employed using manufacturer standard protocols. Amplified signal for intracellular analytes is generated with commercially available antibodies to the target proteins or phospho-proteins. Once these primary antibodies are bound, peroxidase enzyme is bound via secondary antibodies. A tyramide-fluorophore conjugate substrate for peroxidase is then used to amplify the signal. Amplified signal is detected with a regular flow cytometer.

After blood sample collection of the study was completed, the frozen samples were thawed for cell-specific molecular expression analysis by CellPrint Biotechnology, LLC (Cleveland, OH, USA). Acquisition of fluorescence levels was accomplished on a BD Accuri C6 flow cytometer and recorded as median fluorescence intensity (MFI). MFI for each analyte was normalized to the fluorescence minus one (FMO) control to generate a median fluorescence ratio (MFR), which is a quantitative measure of relative protein expression level. MFR = 1 means no detection of an analyte.

Antibodies to 28 proteins were used in the study. The commercially available antibodies were evaluated with proprietary quality controls established by the CellPrint Biotechnology team. The 28 intracellular analytes probed were involved in a spectrum of pathways/functions (Appendix A). The selection of these proteins was based on previous studies [24,25,38,39,40,41,42,69,71,72,73,74].

After the flow cytometric analyses were completed, the raw cytometric data were sent to the clinical investigation site. The clinical team provided the status of each patient as a responder or a non-responder to the statistics team. The statistics team conducted analyses to assess any differences between responders and non-responders, and between before and after lithium treatment.

### 2.4. Normalization of Raw Data

The MFR of each analyte for responders and non-responders, and before and after lithium was “normalized” with fold change/difference (FC). The FC of each analyte was calculated with a formula of $\log_{2}\left(\frac{the\;average\;of\;MFR\;ofNon-Responders\;}{the\;average\;of\;MFR\;of\;Responders\;}\right)$ for comparison between lithium responders and non-responders, or $\log_{2}\left(\frac{the\;average\;of\;MFR\;of\;after\;lithium}{the\;average\;of\;MFR\;of\;before\;lithium\;}\right)$ for before and after lithium comparison. Therefore, a positive value of the log_2_ (FC) is indicative of a higher level in non-responders or after lithium, and a negative value of log_2_ (FC) is indicative of a lower level in non-responders or after lithium.

### 2.5. Statistical Analysis

The statistical method is dependent on the variables being analyzed. Overall, the bivariate data were analyzed with Chi-square or Fisher Exact tests. Continuous variables were analyzed with T-test. Analyses of demographics and clinical characteristics between responders and non-responders were conducted as previously described [45]. Similarly, a ≥50% reduction in Montgomery Asberg Rating Scale (MADRS and/or Young Mania Rating Scale (YMRS) from baseline to the end of study were used to define a responder.

The flow cytometric data of each of the 28 analytes in monocytes and CD4^+^ lymphocytes between responders and non-responders at baseline were analyzed with unpaired t-test, and within group before and after lithium were analyzed with paired t-test. The log_2_ (FC) normalization data and the raw MFR for the analytes were used to study the changes in each analyte before and after lithium. Due to the explorative nature of the study, no adjustment was attempted for multiple comparisons. Data were analyzed using SAS software (SAS version 9.2, SAS Institute Inc., Cary, NC, USA).

## 3. Results

The demographics, historical correlates, and changes in depression and anxiety severities after lithium treatment were described previously [45]. Of the 25 patients who were treated with lithium, twenty-four had at least one post-baseline visit (intent to treat, ITT). Of the 24 patients, 13 were classified as treatment responders (R), and 11 were treatment non-responders (NR).

### 3.1. MFR of Analytes at Baseline: ITT-Responders vs. ITT-Non-Responders

The blood samples of 12 of 13 ITT-responders (ITT-R-baseline) and all 11-ITT non-responders (ITT-NR-baseline) were available for baseline analysis of 28 analytes. The expression levels of these analytes between ITT-R-baseline and ITT-NR-baseline were not significantly different in both monocytes (Table 1) and CD4^+^ lymphocytes (Appendix A). With the exception of a few analytes, a majority of analytes had higher levels in ITT-NR-baseline than in ITT-R-baseline in both cell types although the magnitude of differences varied widely (Figure 1).

### 3.2. MFR of Analytes at Baseline and at the End of Study: Completed Responders vs. Completed Non-Responders

Nine of 12 ITT-R completed the study (C-R) and 4 of 11 ITT-NR completed the study (C-NR). All these completers had blood samples available at both baseline and the end of study (EOS). As with the analysis in the ITT sample, the protein levels between C-R-baseline and C-NR-baseline were not significantly different in both monocytes and CD4^+^ lymphocytes (Appendix A). However, after lithium treatment, there were significant differences in a number of protein levels between C-R-EOS and C-NR-EOS in monocytes (Table 1) and CD4^+^ lymphocytes (Appendix A). The levels of iNOS, NLRP3, phospho-GSK3β, and PGM1were significantly lower in C-NR-EOS than in C-R-EOS in both cell types (Table 1 and Appendix A).

### 3.3. MFR of Analytes before and after Lithium in Completed Responders and Completed Non-Responders

Of the 9 C-R, the expression levels of 28 analytes in monocytes (Table 2) and CD4^+^ lymphocytes (Appendix A) before and after lithium treatment were compared. In the monocytes, PDEB4 (*p* = 0.03) and NRC31 (*p* = 0.03) were significantly increased (Table 2). In the CD4^+^ lymphocytes, PDEB4 was significantly increased (*p* = 0.05) (Appendix A).

Of the 4 C-NR, all 28 analytes before and after lithium treatment were also compared. In the monocytes, NRC31 levels were significantly decreased with lithium (*p* = 0.04) (Table 2). However, in the CD4^+^ lymphocytes, none of the changes in analytes after lithium treatment was significantly different from the levels before lithium (Appendix A).

### 3.4. Fold Change in MFR of Analytes before and after Lithium in Responders and Non-Responders

In C-R and C-NR, the changes in 28 analytes before and after lithium treatment varied widely as manifested with different magnitudes of FC (Figure 2). In monocytes, with the exception of PKC-θ and possibly mTor, all other analytes in C-R were increased with different magnitudes (blue bars in Figure 2a), but only PDEB4 and NRC31 were significantly increased (Table 2). For C-NR, with the exception of phospho-GSK3β, all other analytes were decreased with different magnitudes (brown bars in Figure 2a), but only the change in NRC31 was significantly different (Table 2). The changes in most analytes before and after lithium were in opposite directions between C-R and C-NR.

Similarly, in CD4^+^ lymphocytes, all analytes in C-R were also increased with different magnitude, but only the increase in PDEB4 (*p* = 0.051) was significantly different (blue bars in Figure 2b, Appendix A). In contrast, almost all analytes in C-NR were decreased with different magnitude (brown bars in Figure 2b), with PKA C-α having the largest difference. However, individually, none was significantly different (Appendix A).

## 4. Discussion

In this pilot study, we employed a sensitive enhanced flow cytometric analysis to quantify protein expression changes in specific circulating mononuclear cells from BP I or II disorder lithium responders and non-responders. We found that lithium induced protein expression level changes in CD4^+^ lymphocytes and monocytes of bipolar patients. The direction of changes, upregulation or downregulation, moved in opposite directions depending on whether the patient was responsive or nonresponsive to lithium. Patients responsive to lithium showed a general increase in expression of the 28 analytes tested, whereas nonresponsive patients showed a general decrease in expression.

Although none of the specific analytes showed statistically significant differences between the two groups at baseline, lithium non-responders generally had higher levels of protein expression levels than lithium responders, which is consistent with our previous analysis of 17 intracellular analytes [45]. The upregulation of a majority of proteins in lithium responders and downregulation in non-responders further support that baseline levels of intracellular proteins may determine the treatment responsiveness to lithium. The change in expression levels after lithium in responders suggests that an upregulation may be a component of the lithium response. However, the meaning of the downregulation after lithium treatment in non-responders remains unclear.

Among the 28 analytes in the current study, only GSK3β and phospho-GSK3β were the subject of previous prospective studies [71,73,75], and only one study used lithium monotherapy [71]. These studies indicate that lithium monotherapy or combination therapy with other psychotropics was able to increase phospho-GSK3β levels, but not the total GSK3β levels. However, the relationship between the increase in phospho-GSK3β levels and improvement in the symptom severity of depression or mania was inconsistent. Since we found that lithium responders and non-responders had different changes after lithium treatment (Figure 2), the previous inconsistencies may have arisen because patients in prior studies were not first stratified by lithium responsiveness.

In our current analysis, phospho-GSK3β levels in monocytes were increased after lithium treatment in both responders and non-responders (Figure 2) with a trended significance in responders (Table 2). This result is consistent with previous studies where patients were analyzed without stratification by lithium responsiveness [71,73,75]. Therefore, the outcome of increased phospho-GSK3β with lithium for improving depression, mania, or both remains unclear. Our protein-protein network analysis of 17 analytes [45] indicated that most analytes in the current analysis are in the same protein network as GSK3β. Therefore, the inhibitory effect of phospho-GSK3β could trigger a cascade effect on downstream targets and pathways that are involved in inflammation, energy metabolism, and immune dysfunction [76,77].

One of the downstream targets of GSK3β is NFkB [45,78]. Previous studies have shown that lithium can decrease NFkB expression through inhibition of GSK3β and lithium has an anti-inflammatory effect [79,80,81,82,83]. However, in our current study, phospho-NFkB caused opposite changes in responders and non-responders after lithium in both cell types (Figure 2). Since the NFkB system is involved in immune development, immune response and inflammation [84,85], and cytokines may play a role in the pathophysiology in BD and lithium treatment response [25], it may not be surprising that factors related to inflammation such as NLRP3 and NR3C1 showed larger increases in lithium responders (Table 2). However, the meaning of the increase in NLRP3 and NR3C1 in lithium responders after lithium remains unclear.

The decreased expression levels induced by lithium of multiple analytes in non-responders is consistent with prior reports of the downregulation of genes in non-responders given lithium [15,30]. Since we only measured the change in symptom severity as a “benefit” of lithium treatment, other potential benefits or harm could have been neglected. Antisuicidal effects of lithium without mood stabilization has been reported [86]. Neurons derived from lithium non-responders did respond to lamotrigine [30], but lithium plus lamotrigine in bipolar depression was more effective than lithium alone [4]. The downregulation of some genes and/or proteins may be necessary for lithium non-responders to respond to lamotrigine adjunctive therapy to lithium. The “benefit” from lithium in lithium non-responders is worthy of further exploration.

In our previous study [45], baseline levels of 17 proteins including BCL2, BDNF, calmodulin, Fyn, phospho-Fyn/phospho-Yes, GSK3β, phospho-GSK3αβ, HMGB1, iNOS, IRS2, mTor, NLPR3, PGM1, PKA C-α, PPARγ, phosphorylated nuclear factor NF-kappa-B p65(Ser536) subunit (phospho-RelA), and TPH1 in monocytes and CD4^+^ lymphocytes were measured with *Cellprint*™. The levels of the majority of analytes in lithium responders were lower than in non-responders in both cell types as in the current study, but the level of GSK3β in monocytes was significantly different (*p* = 0.034). Among the 17 analytes assessed in both studies, most FCs of analytes between responders and non-responders in both cell types were larger in the previous study than in the current one. The FCs of GSK3β, phospho-GSK3αβ, and phospho-RelA in monocytes between non-responders and responders were 0.72, 0.47, and 0.73, respectively. In CD4^+^ lymphocytes, the FCs of phospho-GSK3αβ and GSK3β between two groups were 0.57 and 0.53, respectively. However, in the current study, none of the FCs was over 0.4 (Figure 1). The main differences between these two analyses were: (1) the number of the analytes in the current study was 28 versus 17 in the previous one; (2) the analysis of the previous study was performed after completing the analysis of the current study and proprietary improvement of the *CellPrint*^™^.

We also found that the combination of GSK3β and phospho-GSK3αβ levels in monocytes was able to correctly classified 11/11 responders and 5/8 non-responders. The combination of GSK3β, phospho-RelA, TPH1 and PGM1 correctly predicted 10/11 responders and 6/7 non-responders, both with a likelihood of ≥85%. In addition, signaling pathways of BDNF, neurotrophin, prolactin, leptin, and epidermal growth factor/epidermal growth factor receptors were found to be involved in the lithium treatment response. Similarly, in the current study, BDNF, neurotrophin, prolactin, and leptin pathways were involved in lithium response (data not shown).

Taken together, measuring multiple intracellular proteins with high sensitive flow cytometry such as the *CellPrint*^™^ may help us find biomarkers for predicting lithium treatment response in BD. A multiple approach model including clinical phenotypes, omics, neuroimaging, neuropsychological profiles, and neurophysiological characteristics may be necessary to achieve this goal [87].

### Limitations

Although our results are promising, we should be cautious when interpreting the results. The sample size was small, and only four patients in the non-response group had pre- and post—lithium blood collection. Some markers might have significant differences between lithium responders and non-responders, and/or between before and after lithium with a larger sample size. This is the first study using the significantly more sensitive technique of the *CellPrint*^™^ to measure intracellular proteins in CD4^+^ lymphocytes and monocytes in patients with BD. Although more studies are needed to assess the utility of *CellPrint*^™^ in bipolar research, our current and previous analyses [45] along with the use of the *CellPrint*^™^ platform for different diseases [46,47,48,49,50,51,52] suggest that this technology may help the field to elucidate protein biomarkers in blood cells for predicting lithium response and its mechanism.

## 5. Conclusions

The preliminary results of the current study suggest that enhanced flow cytometry can be used to measure multiple intracellular proteins in the peripheral blood cells of patients with BD. Differences between lithium responders and non-responders at baseline may help us find biomarkers for predicting lithium treatment response. Differences between before and after lithium may shed light on the mechanism of lithium response. Large studies are needed to explore the utility of enhanced flow cytometry such as the *CellPrint*^™^ in searching for predictors of lithium treatment response.

## Figures and Tables

**Figure 1 medicina-59-00120-f001:**
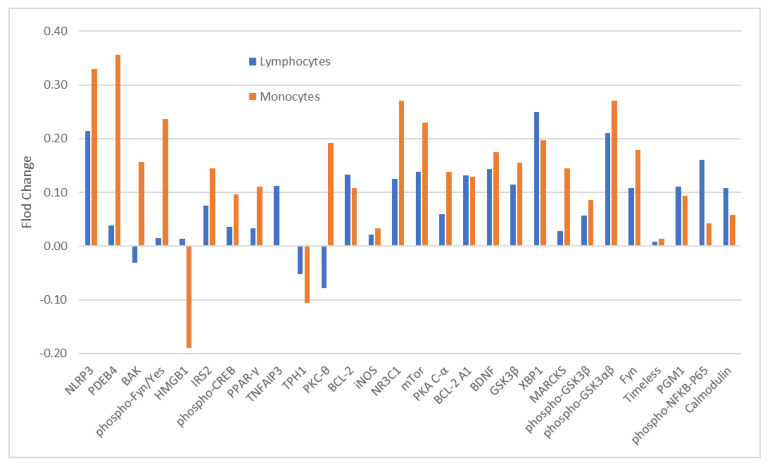
Fold change (FC) of 28 analytes in monocytes and CD4^+^ lymphocytes at baseline between lithium responders and non-responders. Note: FC = log2 (median fluorescent ratio of non-responders/median fluorescent ratio of responders). Positive value is indicative of a higher level of protein expression in lithium non-responders than in lithium responders. Negative value is indicative of a lower level of protein expression in lithium non-responders than in lithium responders. Abbreviations: BAK: BAX, BCL2-Associated × Protein; BCL-2: B-cell lymphoma 2; BCL-2 A1: Bcl-2-related protein A1; BDNF: brain-derived neurotrophic factor; Calmodulin: calciummodulated protein; Fyn: a tyrosine kinase belongs to the Src family of tyrosine kinases including src, fyn, and yes; GSK3*β*: glycogen synthase kinase 3 beta; HMGB1: High mobility group box 1 protein; iNOS: inducible isoform nitric oxide synthases; IRS2: Insulin receptor substrate 2; MARCKS: myristoylated alaninerich C-kinase substrate; mTor: mammalian target of rapamycin; NLRP3: NACHT, LRR and PYD domains-containing protein 3; NR3C1: nuclear receptor subfamily 3, group C, member 1; phospho-CREB: phosphorylated cAMP response element-binding protein (Ser133); phospho-Fyn/Yes: phosphorylated Fyn(Y530)/Yes(Y537); phospho-GSK3*αβ*: phosphorylated glycogen synthase kinase 3 alpha(Tyr279) beta(Tyr216); phospho-GSK 3*β*: phospho-glycogen synthase kinase 3 beta(Tyr216); phospho-NFkB-P65: phosphorylated nuclear factor NF-kappa-B p65(Ser536) subunit; PDEB4: cAMP-specific 3′,5′-cyclic phosphodiesterase 4B; PGM1: phosphoglucomutase 1; PKA C-*α*: protein kinase A catalytic subunit; PKC-θ: protein kinase C theta; PPAR-*γ*: peroxisome proliferator-activated receptor gamma; Timeless: a protein is necessary of proper functioning of circadian rhythm; TNFAIP3: tumor necrosis factor, alpha-induced protein 3; TPH1: tryptophan hydroxylase 1; XBP1: X-box binding protein 1.

**Figure 2 medicina-59-00120-f002:**
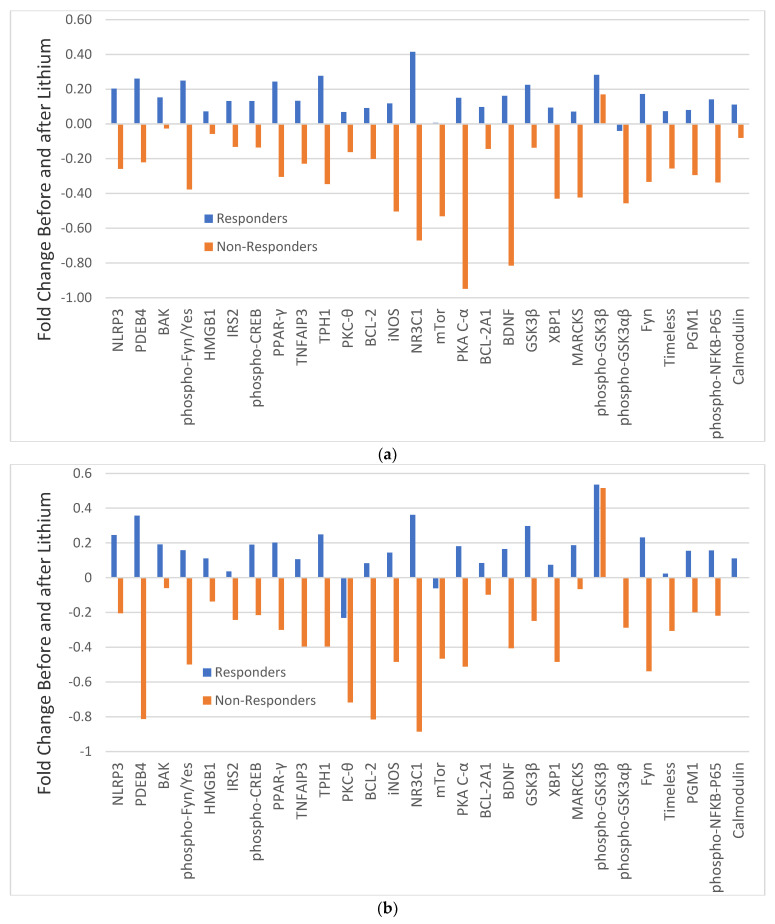
Fold change (FC) of 28 analytes in monocytes (**a**) and CD4^+^ lymphocytes (**b**) in responders (blue bars) and non-responders (brown bars) before and after lithium. Note: FC = log2 (median fluorescent ratio of after lithium/median fluorescent ratio of before lithium Positive value is indicative of a higher level of protein expression after lithium than before lithium. Negative value is indicative of a lower level of protein expression after lithium than before lithium. Abbreviations: BAK: BAX, BCL2-Associated × Protein; BCL-2: B-cell lymphoma 2; BCL-2 A1: Bcl-2-related protein A1; BDNF: brain-derived neurotrophic factor; Calmodulin: calcium-modulated protein; Fyn: a tyrosine kinase belongs to the Src family of tyrosine kinases including src, fyn, and yes; GSK3*β*: glycogen synthase kinase 3 beta; HMGB1: High mobility group box 1 protein; iNOS: inducible isoform nitric oxide synthases; IRS2: Insulin receptor substrate 2; MARCKS: myristoylated alanine-rich C-kinase substrate; mTor: mammalian target of rapamycin; NLRP3: NACHT, LRR and PYD domains-containing protein 3; NR3C1: nuclear receptor subfamily 3, group C, member 1; phospho-CREB: phosphorylated cAMP response element-binding protein (Ser133); phospho-Fyn/Yes: phosphorylated Fyn(Y530)/Yes(Y537); phospho-GSK3*α**β*: phosphorylated glycogen synthase kinase 3 alpha(Tyr279) beta(Tyr216); phospho-GSK 3*β*: phospho-glycogen synthase kinase 3 beta(Tyr216); phospho-NFkB-P65: phosphorylated nuclear factor NF-kappa-B p65(Ser536) subunit; PDEB4: cAMP-specific 3′,5′-cyclic phosphodiesterase 4B; PGM1: phosphoglucomutase 1; PKA C-*α*: protein kinase A catalytic subunit; PKC-θ: protein kinase C theta; PPAR-*γ*: peroxisome proliferator-activated receptor gamma; Timeless: a protein is necessary of proper functioning of circadian rhythm;TNFAIP3: tumor necrosis factor, alpha-induced protein 3; TPH1: tryptophan hydroxylase 1; XBP1: X-box binding protein 1.

**Table 1 medicina-59-00120-t001:** Comparison of protein levels in monocytes between lithium responders and non-responders before and lithium treatment.

	Before Lithium					After Lithium				
Analytes		Intent-to-Treat-Responders (ITT-R)	Intent-to-Treat -Non-Responders (ITT-NR)		ITT-R vs. ITT-NR	Completed-Responders (C-R)		Completed-Non-Responders (C-NR)		C-R vs. C-NR
	*n*	Mean ± SD	*n*	Mean ± SD	*p*-value	*n*	Mean ± SD	*n*	Mean ± SD	*p*-Value
**BAK**	12	23.1 ± 5.2	11	25.3 ± 6.0	0.367	9	26.5 ± 3.0	4	19.6 ± 5.9	0.134
**BCL-2**	11	21.4 ± 4.6	11	23.3 ± 9.0	0.551	9	23.5 ± 3.0	4	13.0 ± 8.4	0.117
**BCL-2 A1**	12	1.4 ± 0.4	11	1.5 ± 0.3	0.380	9	1.5 ± 0.1	4	1.4 ± 0.2	0.298
**BDNF**	12	48.1 ± 12.5	10	54.6 ± 9.3	0.208	9	56.1 ± 8.7	4	36.2 ± 13.3	0.157
**Calmodulin**	12	46.4 ± 11.6	11	48.3 ± 10.3	0.712	9	52.2 ± 8.7	4	41.6 ± 17.5	0.377
**Fyn**	12	34.1 ± 7.5	11	38.4 ± 6.7	0.188	9	40.4 ± 3.9	4	24.3 ± 7.7	0.031
**GSK3** **β**	12	27.5 ± 10.7	10	29.8 ± 10.4	0.625	9	36.6 ± 5.4	4	24.8 ± 5.8	0.028
**HMGB1**	11	26.2 ± 8.9	8	21.3 ± 5.8	0.240	8	27.0 ± 9.8	4	16.4 ± 6.1	0.070
**iNOS**	12	27.9 ± 12.9	11	27.6 ± 6.1	0.944	9	29.3 ± 6.5	4	17.2 ± 5.5	0.021
**IRS2**	12	42.3 ± 13.7	11	46.2 ± 8.7	0.464	9	44.7 ± 10.8	4	38.7 ± 13.8	0.525
**MARCKS**	12	37.0 ± 8.8	11	40.2 ± 10.0	0.440	9	43.1 ± 6.5	4	35.4 ± 16.1	0.477
**mTor**	12	21.5 ± 10.7	11	24.9 ± 7.9	0.433	9	22.1 ± 4.4	4	20.2 ± 7.8	0.719
**NR3C1**	8	7.0 ± 1.5	4	8.5 ± 2.6	0.228	9	39.7 ± 6.5	4	30.5 ± 14.6	0.358
**NLRP3**	12	32.3 ± 10.7	10	41.2 ± 8.4	0.062	9	9.0 ± 1.5	4	4.6 ± 1.1	0.001
**PDEB4**	10	22.4 ± 8.1	5	27.8 ± 9.8	0.308	9	26.5 ± 3.0	3	15.8 ± 10.4	0.172
**phospho-CREB**	12	46.4 ± 9.2	11	48.4 ± 8.2	0.596	9	46.2 ± 6.7	4	34.7 ± 11.1	0.168
**phospho-Fyn Yes**	12	23.9 ± 10.9	11	26.6 ± 6.7	0.517	9	52.8 ± 5.3	4	40.3 ± 15.5	0.256
**phospho-GSK3** **β**	12	4.7 ± 1.4	11	4.5 ± 1.1	0.815	9	27.6 ± 7.3	4	18.8 ± 4.5	0.041
**phospho-GSK3** **αβ**	12	3.0 ± 1.0	11	3.7 ± 1.0	0.146	9	6.6 ± 2.7	4	7.2 ± 4.6	0.826
**Phospho-NFkB-P65**	12	8.5 ± 5.2	11	8.2 ± 2.9	0.861	9	3.2 ± 1.0	4	3.6 ± 1.0	0.552
**PGM1**	12	39.8 ± 10.4	11	41.8 ± 8.0	0.628	9	10.9 ± 2.6	4	7.5 ± 1.2	0.014
**PKA C-** **α**	12	39.5 ± 11.01	11	43.0 ± 8.0	0.437	9	47.9 ± 6.7	4	29.1 ± 11.2	0.130
**PKC-** **θ**	9	5.2 ± 2.0	5	6.1 ± 3.0	0.566	9	4.4 ± 1.0	4	3.7 ± 1.6	0.486
**PPAR-** **γ**	12	39.2 ± 11.3	11	41.8 ± 7.0	0.548	9	24.5 ± 7.0	4	32.4 ± 10.7	0.102
**Timeless**	12	1.79 ± 0.48	11	1.8 ± 0.2	0.936	9	1.8 ± 0.2	4	1.4 ± 0.4	0.209
**TNFAIP3**	12	24.71 ± 11.58	11	24.43 ± 10.78	0.9564	9	46.4 ± 6.1	4	14.8 ± 3.9	0.014
**TPH1**	12	16.75 ± 5.65	11	15.29 ± 2.95	0.4897	9	19.1 ± 3.9	4	12.2 ± 3.4	0.028
**XBP1**	12	1.21 ± 0.31	11	1.39 ± 0.48	0.3027	9	1.3 ± 0.2	4	1.2 ± 0.1	0.130

**Abbreviation:** BAK: BAX, BCL2-Associated × Protein; **BCL-2**: B-cell lymphoma 2; BCL-2 A1: Bcl-2-related protein A1; BDNF: brain-derived neurotrophic factor; Calmodulin: calcium-modulated protein; Fyn: a tyrosine kinase belongs to the Src family of tyrosine kinases including src, fyn, and yes; GSK3*β*: glycogen synthase kinase 3 beta; HMGB1: High mobility group box 1 protein; iNOS: inducible isoform nitric oxide synthases; IRS2: Insulin receptor substrate 2; MARCKS: myristoylated alanine-rich C-kinase substrate; mTor: mammalian target of rapamycin; NLRP3: NACHT, LRR and PYD domains-containing protein 3; NR3C1: nuclear receptor subfamily 3, group C, member 1; phospho-CREB: phosphorylated cAMP response element-binding protein (Ser133); phospho-Fyn/Yes: phosphorylated Fyn(Y530)/Yes(Y537); phospho-GSK3*α**β*: phosphorylated glycogen synthase kinase 3 alpha(Tyr279) beta(Tyr216); phospho-GSK3*β*: phospho-glycogen synthase kinase 3 beta(Tyr216)**;** phospho-NFkB-P65: phosphorylated nuclear factor NF-kappa-B p65(Ser536) subunit; PDEB4: cAMP-specific 3′,5′-cyclic phosphodiesterase 4B; PGM1: phosphoglucomutase 1; PKA C-*α*: protein kinase A catalytic subunit; PKC- θ: protein kinase C theta; PPAR-*γ*: peroxisome proliferator-activated receptor gamma; Timeless: a protein is necessary of proper functioning of circadian rhythm; TNFAIP3: tumor necrosis factor, alpha-induced protein 3; TPH1: tryptophan hydroxylase 1; XBP1: X-box binding protein 1.

**Table 2 medicina-59-00120-t002:** Comparison of protein levels in monocytes of lithium completed responders and non-responders between before and after lithium treatment.

	Completed-Responders					Completed-Non-Responders				
Analytes	Before Lithium		After Lithium		Before vs. After Lithium	Before Lithium		After Lithium		Before vs. After Lithium
	*n*	Mean ± SD	*n*	Mean ± SD	*p*-Value	*n*	Mean ± SD	*n*	Mean ± SD	*p*-Value
**BAK**	9	23.2 ± 5.0	9	26.5 ± 3.2	0.118	4	20.5 ± 2.6	4	19.6 ± 6.8	0.826
**BCL-2**	8	22.1 ± 5.0	8	22.8 ± 2.8	0.742	4	22.9 ± 5.4	4	13.0 ± 9.7	0.124
**BCL-2 A1**	9	1.4 ± 0.3	9	1.5 ± 0.1	0.474	4	1.5 ± 0.2	4	1.4 ± 0.2	0.526
**BDNF**	9	50.1 ± 13.3	9	56.1 ± 9.2	0.277	3	44.3 ± 6.2	3	36.2 ± 16.3	0.461
**Calmodulin**	9	48.4 ± 12.8	9	52.2 ± 9.2	0.474	4	41.6 ± 10.6	4	41.6 ± 20.2	0.999
**Fyn**	9	34.4 ± 8.5	9	40.4 ± 4.1	0.074	4	35.3 ± 7.3	4	24.3 ± 8.9	0.106
**GSK3** **β**	9	29.8 ± 10.3	9	36.6 ± 5.8	0.103	4	29.4 ± 7.6	4	24.8 ± 6.7	0.393
**HMGB1**	8	25.8 ± 9.6	8	27.0 ± 10.5	0.806	4	18.1 ± 3.2	4	16.4 ± 7.1	0.689
**iNOS**	9	26.5 ± 13.8	9	29.3 ± 6.9	0.598	4	24.0 ± 5.8	4	17.2 ± 6.4	0.164
**IRS2**	9	43.7 ± 14.0	9	44.7 ± 11.4	0.860	4	45.7 ± 7.9	4	38.7 ± 16.0	0.456
**MARCKS**	9	37.9 ± 9.9	9	43.1 ± 6.9	0.210	4	27.9 ± 10.2	4	20.2 ± 9.0	0.300
**mTor**	9	23.0 ± 11.5	9	22.1 ± 4.7	0.821	4	37.1 ± 11.6	4	35.4 ± 18.6	0.885
**NLRP3**	9	33.5 ± 11.0	9	39.7 ± 6.9	0.172	3	35.1 ± 7.9	3	23.6 ± 12.0	0.236
**NR3C1**	8	7.0 ± 1.5	8	8.6 ± 1.2	**0.029**	4	8.4 ± 2.6	4	4.6 ± 1.2	**0.037**
**PDEB4**	9	20.7 ± 6.4	9	26.5 ± 3.2	**0.027**	3	30.8 ± 9.4	3	21.1 ± 6.9	0.223
**PGM1**	9	41.5 ± 11.2	9	46.2 ± 7.1	0.303	4	39.8 ± 7.8	4	34.7 ± 12.8	0.518
**phospho-CREB**	9	46.3 ± 9.7	9	52.8 ± 5.6	0.101	4	46.8 ± 6.9	4	40.3 ± 17.9	0.524
**phospho-Fyn/Yes**	9	24.7 ± 10.5	9	27.6 ± 7.7	0.519	4	26.5 ± 5.6	4	18.8 ± 5.2	0.089
**phospho-GSK3** **β**	9	4.5 ± 1.4	9	6.6 ± 2.8	0.073	4	5.1 ± 1.2	4	7.2 ± 5.3	0.458
**phospho-GSK3** **αβ**	9	3.1 ± 1.1	9	3.2 ± 1.1	0.987	4	4.4 ± 1.2	4	3.6 ± 1.1	0.373
**phospho-NFkB-P65**	9	9.7 ± 5.4	9	10.9 ± 2.8	0.590	4	8.7 ± 3.2	4	7.5 ± 1.4	0.510
**PKA C-** **α**	9	42.3 ± 10.9	9	47.9 ± 7.1	0.213	4	38.8 ± 3.2	4	29.1 ± 13.7	0.296
**PKC-** **θ**	9	5.2 ± 2.0	9	4.4 ± 1.0	0.322	4	6.0 ± 3.0	4	3.7 ± 1.9	0.229
**PPAR-** **γ**	9	40.3 ± 11.5	9	46.4 ± 7.4	0.206	4	39.9 ± 9.0	4	32.4 ± 12.3	0.364
**Timeless**	9	1.8 ± 0.5	9	1.8 ± 0.3	0.887	4	1.8 ± 0.1	4	1.4 ± 0.5	0.253
**TNFAIP3**	9	22.8 ± 9.6	9	24.5 ± 6.5	0.659	4	19.5 ± 2.8	4	14.8 ± 4.5	0.128
**TPH1**	9	16.1 ± 5.9	9	19.1 ± 4.2	0.230	4	16.1 ± 4.4	4	12.2 ± 3.9	0.240
**XBP1**	9	1.3 ± 0.3	9	1.3 ± 0.2	0.572	4	1.6 ± 0.7	4	1.2 ± 0.2	0.218

**Abbreviation:** BAK: BAX, BCL2-Associated × Protein; BCL-2: B-cell lymphoma 2; BCL-2 A1: Bcl-2-related protein A1; BDNF: brain-derived neurotrophic factor; Calmodulin: calcium-modulated protein; Fyn: a tyrosine kinase belongs to the Src family of tyrosine kinases including src, fyn, and yes; GSK3*β*: glycogen synthase kinase 3 beta; HMGB1: High mobility group box 1 protein; iNOS: inducible isoform nitric oxide synthases; IRS2: Insulin receptor substrate 2; MARCKS: myristoylated alanine-rich C-kinase substrate; mTor: mammalian target of rapamycin; NLRP3: NACHT, LRR and PYD domains-containing protein 3; NR3C1: nuclear receptor subfamily 3, group C, member 1; phospho-CREB: phosphorylated cAMP response element-binding protein (Ser133); phospho-Fyn/Yes: phosphorylated Fyn(Y530)/Yes(Y537); phospho-GSK3*α**β*: phosphorylated glycogen synthase kinase 3 alpha(Tyr279) beta(Tyr216); phospho-GSK3*β*: phospho-glycogen synthase kinase 3 beta(Tyr216); phospho-NFkB-P65: phosphorylated nuclear factor NF-kappa-B p65(Ser536) subunit; PDEB4: cAMP-specific 3′,5′-cyclic phosphodiesterase 4B; PGM1: phosphoglucomutase 1; PKA C-*α*: protein kinase A catalytic subunit; PKC- θ: protein kinase C theta; PPAR-*γ*: peroxisome proliferator-activated receptor gamma; Timeless: a protein is necessary of proper functioning of circadian rhythm; TNFAIP3: tumor necrosis factor, alpha-induced protein 3; TPH1: tryptophan hydroxylase 1; XBP1: X-box binding protein 1.

## Data Availability

Data are available upon request.

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
