# Peer review of "Divergent Directionality of Immune Cell-Specific Protein Expression between Bipolar Lithium Responders and Non-Responders Revealed by Enhanced Flow Cytometry"

_medicina, 2023, doi:10.3390/medicina59010120_

Round 1

Reviewer 1 Report

This pilot study suggests that using enhanced flow cytometry to measure multiple proteins in the peripheral blood cells of bipolar patients may add important information on the mechanism of lithium treatment response. Lithium induced divergent directionality of protein expression de-pending on the whether the patient was a responder or non-responder.

I've found this study well written and clear, and in my opinion it adds important data to existing literature.

Only a minor point: 

page 7: "The change in expression....suggest" (please correct in suggests )

Author Response

Thanks for reviewer's positive comments. We have corrected the sentence on page 7 as suggested accordingly.  

Reviewer 2 Report

I would like to congratulate the authors for their work in conducting this study and subsequently writing of this manuscript. A few comments are presented below. Original idea and interesting study. Well written and clear for the reader (especially considering a wide target from basic researchers, translational researchers and interested clinical psychiatrists). I have some concerns related to the literature review and the references used and believe there some references could be removed as they have no direct relevance to the subject matter of this particular paper. Statistical analysis overall seems adequate but I would recommend more details (including which statistical software was used for the analysis). In the results section, sub-heading about demographics I must point out that this is quite sparse on detail with no substancial information provided in this paragraph. The authors refer to another publication of their own in which they claim to have described this in detail [Ref. 23]. Either this subheading should be complete or removed althogether as there are no real results or data presenting under this sub-heading.

Author Response

Thanks for reviewer's thoughtful comments. 

I would like to congratulate the authors for their work in conducting this study and subsequently writing of this manuscript. A few comments are presented below. Original idea and interesting study. Well written and clear for the reader (especially considering a wide target from basic researchers, translational researchers and interested clinical psychiatrists).

I have some concerns related to the literature review and the references used and believe there some references could be removed as they have no direct relevance to the subject matter of this particular paper.

Response: Thank you for this observation. We have deleted 5 articles not directly related to this study.

Statistical analysis overall seems adequate but I would recommend more details (including which statistical software was used for the analysis).

Response: We have added. “Data were analyzed using SAS software (SAS version 9.2, SAS Institute Inc., Cary, NC).”

In the results section, sub-heading about demographics I must point out that this is quite sparse on detail with no substantial information provided in this paragraph. The authors refer to another publication of their own in which they claim to have described this in detail [Ref. 23]. Either this subheading should be complete or removed altogether as there are no real results or data presenting under this sub-heading. 

Response: We agreed this sub-title is unnecessary. We have deleted it in the revised version.

Reviewer 3 Report

 MEDICINA publications cater to clinicians, diagnosticians, and researchers, and serve as a forum to discuss the current status of health-related matters and their impact on a global and local scale (citation).

Keeping this aim in mind and to enhance clarity of paper for clinically oriented physicians, I suggest:

Introduction

1. to add a brief history of biomarkers, focused mainly on predictors studied with lithium treatment,

2. to mention some other predictors studied with lithium treatment (clinical predictors, 5-HT binding, epigenetic, methylomic biomarkers), to mention The International Consortium on Lithium Genetics (ConLiGen)

3. to explain the statement in abstract (There is no biomarker to predict lithium response) maybe that lithium is a multitargeting drug, molecular signature of responsivity to lithium could help (?)….

Method

1.to explain why monocytes and CD4+ lymphocytes were chosen

Discussion

1.To discuss the difference between this paper and recently published article (Psychopharmacology Bulletin) based on the same study (number of proteins tested).

2. To add your vision about future - suggestions

  …. to combine omics data including genomics, epigenomes, proteomics, metabolomics and microbiomes for identification and understanding of biological pathways and  networks…..

….development of diagnostic and prognostic algorithms combining individual's genomic information with other predictors (omics, neuroimaging, clinical characteristics) , …. integration of molecular science with that of traditional clinical practice…

Literature

to add some citation dealing the topic  more broadly - Scott J et al, 2019, Pisanu C, 2018, Amare AT, 2017)

Author Response

Thank you for your thoughtful suggestions. Please see point-to-point response. 

MEDICINA publications cater to clinicians, diagnosticians, and researchers, and serve as a forum to discuss the current status of health-related matters and their impact on a global and local scale (citation).

Keeping this aim in mind and to enhance clarity of paper for clinically oriented physicians, I suggest:

Introduction

  1. to add a brief history of biomarkers, focused mainly on predictors studied with lithium treatment,

Response: Thank you for this suggestion. We have added the following paragraph.

The effort of searching biomarkers for predicting lithium treatment response has been for decades. Using blood samples, researchers have investigated potential predictors for lithium response at different levels including genomic [11-14], gene expression [15-20], protein levels [21-25], neurotransmitters, signal transduction and pathways, endocrine systems, cytokines and immune systems, circadian rhythm, and mitochondria [6]. Brain imaging and brain activity-related measures have also been used to study the predictors for lithium treatment response [6, 26, 27}. However, these efforts have yet to produce reliable predictors of lithium responsiveness [28,29] although a large study from International Consortium on Lithium Genetics (ConLi*Gen) found that bipolar patients with low genetic loading for schizophrenia had better response to lithium than those with high genetic loading [14].

  1. to mention some other predictors studied with lithium treatment (clinical predictors, 5-HT binding, epigenetic, methylomic biomarkers), to mention The International Consortium on Lithium Genetics (ConLiGen).

Response: Thanks you for this suggestion. In addition to response to comment #1 (see above), we have added the following sentence.

Demographic and clinical characteristics have been compared between lithium responders and non-responders [3, 7], and some of them have been considered being used as a guidance in clinical practice.

  1. 3. to explain the statement in abstract (There is no biomarker to predict lithium response) …  maybe that lithium is a multitargeting drug, molecular signature of responsivity to lithium could help (?)….

Response: Thank you for this suggestion. The following paragraph has been added.

However, a very small contribution of each gene SNP to a complex disease, a mismatch between mRNA and protein levels [35-37], and the inability to measure post translational protein modifications, such as phosphorylation or methylation events have challenged the usefulness of genomic and transcriptomic approaches in studying biomarkers for predicting treatment response of complex diseases like BD. On the other hand, expressed protein levels and phosphorylation are highly correlated with cellular functions and phenotypes. Proteomic studies are not only likely to help us find different phenotypes based on lithium response and biomarkers, but also help us understand the pathology of BD. However, most plasma/serum-based technologies simultaneously interrogate the averaged productive capabilities of all cells in the body and thus are obligatorily low resolution and low sensitivity. Also, lithium can reach many organs, tissues, neurons, and non-neuronal cells, and act on different genes and pathways [6]. Therefore, it will be difficult to use low sensitive technologies to measure multiple proteins simultaneously in plasma/serum to find predictor(s) for lithium treatment response.

Method

1.to explain why monocytes and CD4+ lymphocytes were chosen.

Response: Thank you for this suggestion. We have added the following paragraphs.

Rationale of Using Monocytes and CD4+ lymphocytes

The comparability of blood and brain have been investigated at different levels. At the DNA methylation level, the brain and blood are highly correlated. At the transcriptome level, whole blood [53, 54] and peripheral blood mononuclear cells [55] had similar gene expression patterns as the brain tissues [56]. A systematic review has shown that neurotropic factors have similar changes in both central nervous system and peripheral blood system [25]. In addition, functional connections between brain and blood cells are through immune cells in the peripheral circulation with the brain [57,58].

Peripheral blood mononuclear cells (PBMC) have been used for genomic, genetic, gene expression, and protein expression studies of lithium treatment response in BD [16, 59-61] as well as diagnostic and pathologic studies of BD [62-67]. Among the blood mononuclear cells, lymphocytes [38, 68, 69] and monocytes [69, 70] have been extensively studied with flow cytometry for different purposes. In addition, collection of blood sample is relatively easy and cheap, and results from the PBMCs can be easily applied to routine clinical practice. Therefore, we chose monocytes and CD4+ lymphocytes as reporters of flow cytometry analysis in the current study.

Discussion

1.To discuss the difference between this paper and recently published article (Psychopharmacology Bulletin) based on the same study (number of proteins tested).

Response: Thanks you for this suggestion. We have added the following paragraphs.

In our previous study [45], baseline levels of 17 proteins including BCL2, BDNF, calmodulin, Fyn, phospho-Fyn/phospho-Yes, GSK3β, phospho-GSK3αβ, HMGB1, iNOS, IRS2, mTor, NLPR3, PGM1, PKA C-α, PPARγ, phosphorylated nuclear factor NF-kappa-B p65(Ser536) subunit (phospho-RelA), and TPH1 in monocytes and CD4+ lymphocytes were measured with Cellprint™. The levels of the majority of analytes in lithium responders were lower than in non-responders in both cell types as in the current study, but the level of GSK3β in monocytes was significantly different (p=0.034). Among the 17 analytes assessed in both studies, most FCs of analytes between responders and non-responders in both cell types were larger in the previous study than in the current one. The FCs of GSK3β, phospho-GSK3αβ, and phospho-RelA in monocytes between non-responders and responders were 0.72, 0.47, and 0.73, respectively. In CD4+ lymphocytes, the FCs of phospho-GSK3αβ and GSK3β between two groups were 0.57 and 0.53, respectively. However, in the current study, none of the FCs was over 0.4 (Figure 1). The main differences between these two analyses were: 1) the number of the analytes in the current study was 28 versus 17 in the previous one; 2) the analysis of the previous study was performed after completing the analysis of the current study and proprietary improvement of the CellPrint.

We also found that the combination of GSK3β and phospho-GSK3αβ levels in monocytes was able to correctly classified 11/11 responders and 5/8 non-responders. The combination of GSK3β, phospho-RelA, TPH1 and PGM1 correctly predicted 10/11 responders and 6/7 non-responders, both with a likelihood of ≥ 85%. In addition, signaling pathways of BDNF, neurotrophin, prolactin, leptin, and epidermal growth factor/epidermal growth factor receptors were found to be involved in the lithium treatment response. Similarly, in the current study, BDNF, neurotrophin, prolactin, and leptin pathways were involved in lithium response (data not shown).

  1. To add your vision about future - suggestions

  …. to combine omics data including genomics, epigenomes, proteomics, metabolomics and microbiomes for identification and understanding of biological pathways and  networks…..

….development of diagnostic and prognostic algorithms combining individual's genomic information with other predictors (omics, neuroimaging, clinical characteristics) , …. integration of molecular science with that of traditional clinical practice…

Response: Thank you for this suggestion. However, because we are still in early phase of precision psychiatry, it is a complex and difficult topic to address. We decided not to extend this discussion. We did add the following sentence.

Taken together, measuring multiple intracellular proteins with high sensitive flow cytometry like the CellPrint may help us find biomarkers for predicting lithium treatment response in BD. A multiple approach model including clinical phenotypes, omics, neuroimaging, neuropsychological profiles, and neurophysiological characteristics may be necessary to achieve this goal [87].

Literature

to add some citation dealing the topic  more broadly - Scott J et al, 2019, Pisanu C, 2018, Amare AT, 2017)

Response:  These articles have been added in the revised version.